# Discovery of 3-(1-Amino-2-phenoxyethylidene)-6-methyl-2*H*-pyran-2,4(3*H*)-dione Derivatives as Novel Herbicidal Leads

Chao-Chao Wang [1], Ke Chen [2], Na Li [1], Xue-Kun Wang [1], Shi-Ben Wang [1], Pan Li [1], Xue-Wen Hua [3], Kang Lei [1,*] and Lu-Sha Ji [1]

1 School of Pharmaceutical Sciences, Liaocheng University, Liaocheng 252000, China
2 Department of Bioscience and Biotechnology, The University of Suwon, Whasung 18323, Republic of Korea
3 College of Agricultural Science and Engineering, Liaocheng University, Liaocheng 252000, China
* Correspondence: leikang@lcu.edu.cn

**Abstract:** Natural products are one of the resources for discovering novel pesticide leads. Here, by molecular hybridization between the natural enamino diketone skeleton and the reported herbicide lead compound **I**, a series of 3-(1-aminoethylidene)-6-methyl-2*H*-pyran-2,4(3*H*)-dione derivatives (**APD**) were rationally designed, synthesized and tested for herbicidal activity in a greenhouse. The bioassay results showed that most of the target compounds possessed good herbicidal activity under pre-emergence conditions, of which the analog **APD-II-15** displayed good pre-emergent herbicidal activity against *Abutilon theophrasti Medicus*, *Amaranthus retroflexus L.*, *Echinochloa crus-galli*, *Eragrostis curvula (Schrad.) Nees*, *Avena fatua L.*, *Cyperus difformis L.*, *Chenopodium album L.*, *Ixeris denticulata*, *Plantago asiatica L.*, *Capsella bursa-pastoris (Linn.) Medic* and *Flaveria bidentis (L.) Kuntze* with >60% inhibition even at a dosage of 187.5 g ha$^{-1}$, and displayed good crop safety for wheat, soybean, millet and sorghum at a dosage of 375 g ha$^{-1}$. The preliminary study of the molecular mode of action by RNA sequencing suggested that a growth inhibition of weeds by **APD-II-15** might result from the disruptions of carbon metabolism and formation of a cytoskeleton. The present work indicated that **APD-II-15** might be used as a novel herbicidal lead compound for further optimization.

**Keywords:** enamino diketone; pyran-2,4(3*H*)-dione; synthesis; herbicide; RNA sequencing

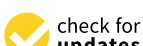



## 1. Introduction

Weeds are the biggest competitor of crops, leading to enormous yield losses in agriculture. It is estimated that if these were left untreated, weeds would reduce global crop yields by 34% [1]. Herbicides, as the efficient weed control tool, play a very important role in protecting crops, increasing yields, and reducing labor costs in agriculture. However, an inevitable problem associated with the overuse of the same herbicide or several herbicides with the same mechanism of action is the evolvement of herbicide resistance [2–5]. To overcome this problem, developing efficient herbicides with novel structures or novel modes of action is necessary [6–8]. Biologically active natural products (NPs) are often served as the lead structures for novel herbicide discovery. To date, various NP-derived herbicides have been extensively developed and brought to the marketplace. For example, glufosinate, served as a commercial glutamine synthetase (GS) inhibitor, is derived from bialaphos obtained from fermentation cultures of the actinomycete *Streptomyces hygroscopis* [9]; mesotrione, served as a commercial 4-hydroxyphenylpyruvate dioxygenase (HPPD) inhibitor, is derived from the allelochemical leptospermone [10]. Furthermore, the introduction of a natural active skeleton is also an efficient method for novel agrichemical discovery [11,12]. For example, Yang et al. introduced the natural active quinazoline-2,4-dione skeleton into triketone HPPD-inhibiting herbicides and successfully developed quinotrione [13,14].

The enamino diketone skeleton is a basic structural feature that can be found in some natural products (Figure 1A) [15,16]. Over the last few decades, compounds with an enamino diketone skeleton (Figure 1B) have played an important role in pharmaceutical and agrochemical fields due to their antimicrobial, [17,18] antibacterial, [19–21] anticancer, [22–26] herbicidal, [27–29] antifungal [30–32] and anti-toxoplasma gondii activity [33]. Such fascinating structural and biological properties of the enamino diketone skeleton enable its application in the development of novel pesticides.

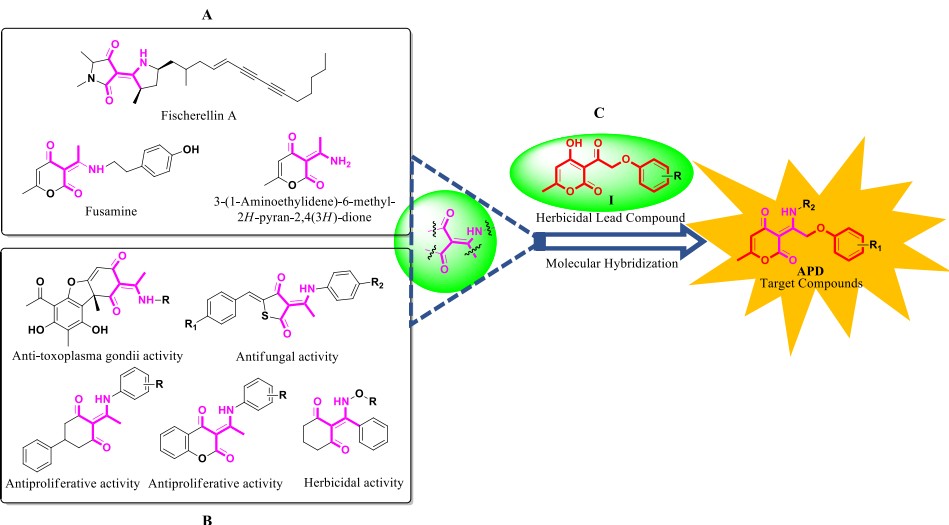

**Figure 1.** Design of target compound **APD** based on the molecular hybridization strategy. (**A**) natural product containing enamino diketone skeleton; (**B**) bioactive molecular containing enamino diketone skeleton; (**C**) the previous reported herbicidal lead compound I.

Recently, during our search for novel compounds that have potential use as herbicides, we have developed a promising herbicide lead: structure **I** (Figure 1C) [34]. It is easy to find that compound **I** has similar diketone moiety compared to the enamino diketone derivative. We envisaged that changing the carbonyl group on the acyl group of compound **I** to an imine and then isomerizing to enamine form can help construct compounds containing the natural enamino diketone skeleton, which should possess good herbicidal activity (Figure 1). Therefore, thirty-three 3-(1-aminoethylidene)-6-methyl-2*H*-pyran-2,4(3*H*)- dione derivatives (**APD**) were designed and synthesized, and their herbicidal activities were evaluated. Herein, we report the synthesis and structure–activity relationship (SAR) of **APD** together with the molecular mode of action study results.

## 2. Materials and Methods

### 2.1. General

In most cases, the reagents and solvents purchased from Energy Chemical or Tokyo Chemical Industry were analytical grade and used without further purification. Column chromatography purification was carried out using silica gel column chromatography (silica gel 200–300 mesh) (Qingdao Makall Group Co., Ltd., Qingdao, China). $^1$H and $^{13}$C NMR spectrum were obtained at 400 MHz and 100 MHz, respectively, using an AV-400 spectrometer (Bruker, Billerica, MA, USA) in CDCl$_3$ or DMSO-d$_6$ solution with tetramethylsilane (TMS) as the internal standard. The chemical shifts are reported as δ values relative to TMS. High-resolution mass spectra were conducted using an Ionspec 7.0 T spectrometer (Varian, Palo Alto, CA, USA) by the electrospray ionization fourier transform ion cyclotron resonance (ESI-FTICR) technique. The crystal structure was determined on a Saturn 724 CCD area-detector diffractometer (Rigaku, Tokyo, Japan).

### 2.2. Chemical Synthesis Procedures

The synthetic pathway used to prepare the target compounds **APD-I** and **APD-II** is outlined in Scheme 1. The yields were not optimized.

**Scheme 1.** Synthetic route for preparing target compounds **APD-I** and **APD-II**. Reagent and conditions: (a) $K_2CO_3$ DMF, 50 °C, 12 h; (b) NaOH, MeOH/$H_2O$, 70 °C, 2 h; (c) oxalyl chloride, DMF, DCM, 25 °C, 12 h; (d) $Et_3N$, DCM, 25 °C, 12 h; (e) $Et_3N$, KCN, 18-crown-6, DCM, 25 °C, 72 h; (f) $Et_3N$, EtOH, 80 °C, 12 h.

2.2.1. General Procedure for Preparing 4-hydroxyl-3-acetyl-pyran-2-one Derivatives 7

The 4-hydroxyl-3-acetyl-pyran-2-one derivatives **7** were prepared according to our previously reported method [34]. The appropriate phenol **1** (0.1 mol) and potassium carbonate (0.2 mol) were added successively to *N, N*-dimethylformamide (250 mL) in a 500 mL flask with stirring, and the mixture was heated to 50 °C for 1 h. Methyl chloroacetate **2** (0.1 mol) was added to the mixture, and the suspension was stirred at 50 °C for 12 h. Then, the mixture was cooled to room temperature, poured into water, and stirred for another 0.5 h. To the solution was added 300 mL of ethyl acetate; the ethyl acetate layer was separated and concentrated by rotary evaporation to give crude intermediate **3**.

Subsequently, 200 mL of methanol were added to dissolve the crude intermediate **3**, and lithium hydroxide (0.5 mol) and water (10 mL) were added to the solution with stirring. The solution was heated to 50 °C for 12 h. After completion of the reaction, 150 mL of methanol was removed by rotary evaporation, and the residue was poured into 200 mL of ice water. The solution was acidified by a hydrochloric acid solution (6 M) to pH =1. The resulting solid was collected by filtration, washed with water and dried in a vacuum to afford phenoxyacetic acid **4**.

Phenoxyacetic acid **4** (10 mmol) was dissolved in dichloromethane (100 mL) in a 250 mL flask; oxalyl chloride (20 mmol) and *N, N*-dimethylformamide (one drop) were added. The mixture was stirred at room temperature for 12 h. After completion of the reaction, dichloromethane was removed by rotary evaporation to provide phenoxyacetyl chloride **5**, which was used in the next step without purification.

Both 4-hydroxy-6-methyl-2*H*-pyran-2-one **6** (5 mmol) and 50 mL of dichloromethane were added to a 100 mL flask, and the solution was cooled to 0 °C. Phenoxyacetyl chloride **5** was added to the solution and stirred for another 30 min; then, triethylamine (7.5 mmol) and

4-dimethylaminopyridine (0.5 mmol) were added successively to the mixture at 0 °C, and the solution was stirred for 12 h. After completion of the reaction, an aqueous hydrochloric acid solution (1 M, 20 mL) was added. The dichloromethane layer was separated and washed with $H_2O$, saturated sodium chloride solution, dried by anhydrous sodium sulfate, concentrated by rotary evaporation, and the residue was dissolved in dichloromethane (20 mL) and added to triethylamine (5 mmol), 18-crown-6 (0.2 mmol) and potassium cyanide (1 mmol). The mixture was stirred at room temperature for 72 h. The mixture was poured into water and extracted with dichloromethane. The combined organic phase was dried with anhydrous sodium sulfate, filtered and removed by rotary evaporation. The residue was scratched from ethyl acetate and ethanol (1/1 by volume) to give intermediate **7** as a solid.

2.2.2. General Preparation of 3-(1-aminoethylidene)-6-methyl-2H-pyran-2,4(3H)-dione Derivatives (Compounds APD-I-1 to APD-I-14 and APD-II-1 to APD-II-19)

A measure of 4-Hydroxyl-3-acetyl-pyran-2-one derivative **7** (2.0 mmol) was dissolved in 20 mL of ethanol, followed by the addition of $Et_3N$ (3.0 mmol) and the corresponding amine (2.2 mmol), and the mixture was refluxed at 80 °C for 12 h. After cooling, the precipitation was selected to give target compounds **APD-I-1** to **APD-I-14** and **APD-II-1** to **APD-II-19**.

Compound 6-Methyl-3-(1-(methylamino)-2-phenoxyethylidene)-2*H*-pyran-2,4(3*H*)-dione (**APD-I-1**): light yellow solid; yield 77.2%; $^1$H NMR (400 MHz, CDCl$_3$) $\delta$: 14.30 (s, 1H), 7.32–7.28 (m, 2H), 7.08–6.91 (m, 3H), 5.74 (s, 1H), 5.68 (s, 2H), 3.34 (d, $J$ = 4.0 Hz, 3H), 2.15 (s, 3H); $^{13}$C NMR (100 MHz, CDCl$_3$) $\delta$: 185.0, 173.0, 163.4, 163.1, 157.4, 129.8, 122.0, 114.5, 107.4, 96.2, 63.2, 32.8, 19.9; HRMS: calcd for $C_{15}H_{16}NO_4^+$ [M + H]$^+$ 274.1074, found 274.1080; calcd for $C_{15}H_{15}NNaO_4^+$ [M + Na]$^+$ 296.0893, found 296.0901.

Compound 6-Methyl-3-(1-(methylamino)-2-(*o*-tolyloxy)ethylidene)-2*H*-pyran-2,4(3*H*)-dione (**APD-I-2**): light yellow solid; yield 73.0%; $^1$H NMR (400 MHz, CDCl$_3$) $\delta$: 14.29 (bs, 1H), 7.18 (t, $J$ = 7.7 Hz, 1H), 6.88–6.73 (m, 3H), 5.74 (d, $J$ = 0.8 Hz, 1H), 5.64 (s, 2H), 3.34 (d, $J$ = 4.0 Hz, 3H), 2.33 (s, 3H), 2.15 (s, 3H); $^{13}$C NMR (100 MHz, CDCl$_3$) $\delta$: 185.0, 173.1, 163.4, 163.1, 157.4, 139.9, 129.5, 122.8, 115.4, 111.33, 107.4, 96.2, 63.3, 32.7, 21.5, 19.9; HRMS: calcd for $C_{16}H_{18}NO_4^+$ [M + H]$^+$ 288.1230, found 288.1238; calcd for $C_{16}H_{17}NNaO_4^+$ [M + Na]$^+$ 310.1050, found 310.1060.

Compound 6-Methyl-3-(1-(methylamino)-2-(*m*-tolyloxy)ethylidene)-2*H*-pyran-2,4(3*H*)-dione (**APD-I-3**): light yellow solid; yield 71.3%; $^1$H NMR (500 MHz, CDCl$_3$) $\delta$: 14.28 (s, 1H), 7.18 (t, $J$ = 7.8 Hz, 1H), 6.85–6.74 (m, 3H), 5.73 (d, $J$ = 0.7 Hz, 1H), 5.64 (s, 2H), 3.34 (d, $J$ = 5.2 Hz, 3H), 2.33 (s, 3H), 2.14 (s, 3H); $^{13}$C NMR (125 MHz, CDCl$_3$) $\delta$: 185.0, 173.1, 163.4, 163.1, 157.5, 139.9, 129.5, 122.9, 115.4, 111.4, 107.4, 96.2, 63.3, 32.7, 21.5, 19.8; HRMS: calcd for $C_{16}H_{18}NO_4^+$ [M + H]$^+$ 288.1230, found 288.1241; calcd for $C_{16}H_{17}NNaO_4^+$ [M + Na]$^+$ 310.1050, found 310.1062.

Compound 6-Methyl-3-(1-(methylamino)-2-(p-tolyloxy)ethylidene)-2*H*-pyran-2,4(3*H*)-dione (**APD-I-4**): light yellow solid; yield 79.7%; $^1$H NMR (400 MHz, CDCl$_3$) $\delta$: 14.28 (bs, 1H), 7.10 (d, $J$ = 8.4 Hz, 2H), 6.93–6.82 (m, 2H), 5.73 (s, 1H), 5.64 (s, 2H), 3.34 (d, $J$ = 5.3 Hz, 3H), 2.29 (s, 3H), 2.14 (s, 3H); $^{13}$C NMR (100 MHz, CDCl$_3$) $\delta$: 185.0, 173.1, 163.4, 163.1, 157.4, 139.9, 129.5, 122.8, 115.4, 111.33, 107.4, 96.2, 63.3, 32.7, 21.5, 19.9; HRMS: calcd for $C_{16}H_{18}NO_4^+$ [M + H]$^+$ 288.1230, found 288.1237; calcd for $C_{16}H_{17}NNaO_4^+$ [M + Na]$^+$ 310.1050, found 310.1058.

Compound 3-(2-(2-Chlorophenoxy)-1-(methylamino)ethylidene)-6-methyl-2*H*-pyran-2,4(3*H*)-dione (**APD-I-5**): light yellow solid; yield 71.4%; $^1$H NMR (400 MHz, CDCl$_3$) $\delta$: 14.38 (bs, 1H), 7.40–7.37 (m, 1H), 7.25–7.21 (m, 1H), 7.14 (dd, $J$ = 8.3, 1.2 Hz, 1H), 6.97 (td, $J$ = 7.7, 1.3 Hz, 1H), 5.77 (s, 2H), 5.75 (s, 1H), 3.45 (d, $J$ = 4.0 Hz, 3H), 2.15 (s, 3H); $^{13}$C NMR (100 MHz, CDCl$_3$) $\delta$: 185.0, 172.2, 163.6, 163.1, 152.9, 130.6, 128.2, 122.8, 114.1, 107.5, 96.2, 64.1, 33.1, 19.9; HRMS: calcd for $C_{15}H_{15}ClNO_4^+$ [M + H]$^+$ 308.0684, found 308.0695; calcd for $C_{15}H_{14}ClNNaO_4^+$ [M + Na]$^+$ 330.0504, found 330.0514.

Compound 3-(2-(3-Chlorophenoxy)-1-(methylamino)ethylidene)-6-methyl-2*H*-pyran-2,4(3*H*)-dione (**APD-I-6**): light yellow solid; yield 73.0%; $^1$H NMR (400 MHz, CDCl$_3$) $\delta$: 14.31 (bs, 1H), 7.23 (t, *J* = 8.1 Hz, 1H), 7.01–7.00 (m, 2H), 6.91–6.86 (m, 1H), 5.75 (s, 1H), 5.65 (s, 2H), 3.33 (d, *J* = 4.0 Hz, 3H), 2.15 (s, 3H); $^{13}$C NMR (100 MHz, CDCl$_3$) $\delta$: 185.0, 172.1, 163.4, 163.3, 158.1, 135.2, 130.6, 122.3, 115.5, 112.7, 107.4, 96.3, 63.3, 32.6, 19.9; HRMS: calcd for C$_{15}$H$_{15}$ClNO$_4$$^+$ [M + H]$^+$ 308.0684, found 308.0695; calcd for C$_{15}$H$_{14}$ClNNaO$_4$$^+$ [M + Na]$^+$ 330.0504, found 330.0514.

Compound 3-(2-(4-Chlorophenoxy)-1-(methylamino)ethylidene)-6-methyl-2*H*-pyran-2,4(3*H*)-dione (**APD-I-7**): light yellow solid; yield 78.1%; $^1$H NMR (400 MHz, CDCl$_3$) $\delta$: 14.28 (bs, 1H), 7.10 (d, *J* = 8.4 Hz, 2H), 6.87 (d, *J* = 8.6 Hz, 2H), 5.73 (s, 1H), 5.64 (s, 2H), 3.34 (d, *J* = 4.0 Hz, 3H), 2.15 (s, 3H); $^{13}$C NMR (100 MHz, CDCl$_3$) $\delta$: 185.0, 173.2, 163.4, 163.1, 155.6, 131.1, 127.2, 126.6, 121.7, 111.3, 107.4, 96.2, 63.4, 32.6, 19.9; HRMS: calcd for C$_{15}$H$_{15}$ClNO$_4$$^+$ [M + H]$^+$ 308.0684, found 308.0690; calcd for C$_{15}$H$_{14}$ClNNaO$_4$$^+$ [M + Na]$^+$ 330.0504, found 330.0510.

Compound 3-(2-(2,3-Dichlorophenoxy)-1-(methylamino)ethylidene)-6-methyl-2*H*-pyran-2,4(3*H*)-dione (**APD-I-8**): white solid; yield 70.0%; $^1$H NMR (400 MHz, CDCl$_3$) $\delta$: 14.39 (bs, 1H), 7.20–7.11 (m, 2H), 7.07 (dd, *J* = 7.8, 1.8 Hz, 1H), 5.79 (s, 2H), 5.76 (s, 1H), 3.44 (d, *J* = 4.0 Hz, 3H), 2.16 (s, 3H); $^{13}$C NMR (100 MHz, CDCl$_3$) $\delta$: 185.0, 171.7, 163.6, 163.2, 154.3, 134.2, 127.8, 123.7, 122.0, 112.0, 107.5, 96.2, 64.2, 33.0, 19.9; HRMS: calcd for C$_{15}$H$_{13}$Cl$_2$NNaO$_4$$^+$ [M + Na]$^+$ 364.0114, found 364.0125.

Compound 3-(2-(2,4-Chlorophenoxy)-1-(methylamino)ethylidene)-6-methyl-2*H*-pyran-2,4(3*H*)-dione (**APD-I-9**): white; yield 79.3%; $^1$H NMR (400 MHz, CDCl$_3$) $\delta$: 14.38 (bs, 1H), 7.38 (d, *J* = 2.5 Hz, 1H), 7.21 (dd, *J* = 8.8, 2.5 Hz, 1H), 7.11 (d, *J* = 8.8 Hz, 1H), 5.75 (bs, 3H), 3.44 (d, *J* = 4.0 Hz, 3H), 2.15 (s, 3H); $^{13}$C NMR (100 MHz, CDCl$_3$) $\delta$: 185.0, 171.6, 163.6, 163.2, 151.8, 130.3, 128.1, 127.4, 123.8, 115.2, 107.5, 96.2, 64.2, 32.9, 19.9; HRMS: calcd for C$_{15}$H$_{14}$Cl$_2$NO$_4$$^+$ [M + H]$^+$ 342.0294, found 342.0304; calcd for C$_{15}$H$_{13}$Cl$_2$NNaO$_4$$^+$ [M + Na]$^+$ 364.0114, found 364.0126.

Compound 3-(2-(2,5-Chlorophenoxy)-1-(methylamino)ethylidene)-6-methyl-2*H*-pyran-2,4(3*H*)-dione (**APD-I-10**): white solid; yield 79.0%; $^1$H NMR (400 MHz, CDCl$_3$) $\delta$: 14.38 (bs, 1H), 7.30 (d, *J* = 8.5 Hz, 1H), 7.17 (d, *J* = 2.2 Hz, 1H), 6.97 (dd, *J* = 8.5, 2.2 Hz, 1H), 5.76 (s, 1H), 5.72 (s, 2H), 3.42 (d, *J* = 4.0 Hz, 3H), 2.16 (s, 3H); $^{13}$C NMR (100 MHz, CDCl$_3$) $\delta$: 185.0, 171.2, 163.5, 163.3, 153.4, 133.5, 131.0, 122.9, 121.4, 114.8, 107.5, 96.3, 64.1, 32.8, 19.9; HRMS: calcd for C$_{15}$H$_{14}$Cl$_2$NO$_4$$^+$ [M + H]$^+$ 342.0294, found 342.0306; calcd for C$_{15}$H$_{13}$Cl$_2$NNaO$_4$$^+$ [M + Na]$^+$ 364.0114, found 364.0124.

Compound 3-(2-(3,5-Chlorophenoxy)-1-(methylamino)ethylidene)-6-methyl-2*H*-pyran-2,4(3*H*)-dione (**APD-I-11**): white solid; yield 72.1%; $^1$H NMR (400 MHz, CDCl$_3$) $\delta$: 14.32 (bs, 1H), 7.03 (s, 1H), 6.91 (d, *J* = 1.6 Hz, 2H), 5.75 (s, 1H), 5.62 (s, 2H), 3.32 (d, *J* = 4.0 Hz, 3H), 2.15 (s, 3H); $^{13}$C NMR (100 MHz, CDCl$_3$) $\delta$: 185.1, 171.2, 163.4, 163.3, 158.4, 135.7, 122.5, 113.8, 107.4, 96.3, 63.2, 32.4, 19.9; HRMS: calcd for C$_{15}$H$_{14}$Cl$_2$NO$_4$$^+$ [M + H]$^+$ 342.0294, found 342.0306; calcd for C$_{15}$H$_{13}$Cl$_2$NNaO$_4$$^+$ [M + Na]$^+$ 364.0114, found 364.0125.

Compound 3-(2-(4-Chloro-2-methylphenoxy)-1-(methylamino)ethylidene)-6-methyl-2*H*-pyran-2,4(3*H*)-dione (**APD-I-12**): light yellow solid; yield 76.6%; $^1$H NMR (400 MHz, CDCl$_3$) $\delta$: 14.34 (bs, 1H), 7.17–7.09 (m, 2H), 6.87 (d, *J* = 9.4 Hz, 1H), 5.75 (s, 1H), 5.65 (s, 2H), 3.35 (d, *J* = 4.0 Hz, 3H), 2.20 (s, 3H), 2.15 (s, 3H); $^{13}$C NMR (100 MHz, CDCl$_3$) $\delta$: 185.1, 172.6, 163.4, 163.2, 154.3, 130.8, 128.5, 126.8, 126.5, 112.6, 107.4, 96.2, 63.5, 32.5, 19.9, 16.3; HRMS: calcd for C$_{16}$H$_{17}$ClNO$_4$$^+$ [M + H]$^+$ 322.0841, found 322.0849; calcd for C$_{16}$H$_{16}$ClNNaO$_4$$^+$ [M + Na]$^+$ 344.0660, found 344.0669.

Compound 3-(2-(4-Bromo-2-chlorophenoxy)-1-(methylamino)ethylidene)-6-methyl-2*H*-pyran-2,4(3*H*)-dione (**APD-I-13**): white solid; yield 81.3%; $^1$H NMR (400 MHz, CDCl$_3$) $\delta$: 14.38 (bs, 1H), 7.52 (d, *J* = 2.4 Hz, 1H), 7.35 (dd, *J* = 8.8, 2.4 Hz, 1H), 7.05 (d, *J* = 8.8 Hz, 1H), 5.75 (bs, 3H), 3.43 (d, *J* = 4.0 Hz, 3H), 2.15 (s, 3H); $^{13}$C NMR (100 MHz, CDCl$_3$) $\delta$: 185.0, 171.6, 163.6, 163.2, 152.3, 133.0, 131.0, 124.1, 115.6, 114.3, 107.5, 96.2, 64.1, 32.9, 19.9; HRMS: calcd for C$_{15}$H$_{14}$BrClNO$_4$$^+$ [M + H]$^+$ 385.9789, found 385.9801; calcd for C$_{15}$H$_{13}$BrClNNaO$_4$$^+$ [M + Na]$^+$ 407.9609, found 407.9620.

Compound 3-(2-(2-Chloro-4-fluorophenoxy)-1-(methylamino)ethylidene)-6-methyl-2$H$-pyran-2,4(3$H$)-dione (**APD-I-14**): white solid; yield 77.7%; $^1$H NMR (400 MHz, CDCl$_3$) $\delta$: 14.37 (bs, 1H), 7.18–7.12 (m, 2H), 7.01–6.92 (m, 1H), 5.75 (s, 1H), 5.72 (s, 2H), 3.45 (d, $J$ = 8.0 Hz, 3H), 2.15 (s, 3H); $^{13}$C NMR (100 MHz, CDCl$_3$) $\delta$: 185.0, 171.7, 157.3 (d, $J$ = 244.1 Hz), 149.6 (d, $J$ = 3.0 Hz), 123.8 (d, $J$ = 10.5 Hz), 117.8 (d, $J$ = 26.1 Hz), 115.6 (d, $J$ = 8.8 Hz), 114.6 (d, $J$ = 22.7 Hz), 107.5, 96.2, 64.6, 32.9, 19.9; HRMS: calcd for C$_{15}$H$_{14}$ClFNO$_4$$^+$ [M + H]$^+$ 326.0590, found 326.0600; calcd for C$_{15}$H$_{13}$ClFNNaO$_4$$^+$ [M + Na]$^+$ 348.0409, found 348.0419.

Compound 3-(2-(2-Chloro-4-fluorophenoxy)-1-(ethylamino)ethylidene)-6-methyl-2$H$-pyran-2,4(3$H$)-dione (**APD-II-1**): white solid; yield 67.5%; $^1$H NMR (400 MHz, CDCl$_3$) $\delta$: 14.39 (s, 1H), 7.19–7.10 (m, 2H), 7.01–6.90 (m, 1H), 5.74 (s, 1H), 5.68 (s, 2H), 3.91–3.81 (m, 2H), 2.15 (s, 3H), 1.42 (t, $J$ = 7.3 Hz, 3H); $^{13}$C NMR (100 MHz, CDCl$_3$) $\delta$: 185.0, 170.3, 163.6, 163.1, 157.3 (d, $J$ = 244.1 Hz), 149.7, 123.8 (d, $J$ = 10.5 Hz), 117.7 (d, $J$ = 26.2 Hz), 115.7 (d, $J$ = 8.8 Hz), 114.6 (d, $J$ = 22.6 Hz), 107.5, 95.8, 64.8, 41.2, 19.9, 15.1; HRMS: calcd for C$_{16}$H$_{16}$ClFNO$_4$$^+$ [M + H]$^+$ 340.0746, found 340.0759; calcd for C$_{16}$H$_{15}$ClFNNaO$_4$$^+$ [M + Na]$^+$ 362.0566, found 362.0576.

Compound 3-(2-(2-Chloro-4-fluorophenoxy)-1-(propylamino)ethylidene)-6-methyl-2$H$-pyran-2,4(3$H$)-dione (**APD-II-2**): light yellow solid; yield 72.0%; $^1$H NMR (400 MHz, CDCl$_3$) $\delta$: 14.45 (s, 1H), 7.18–7.11 (m, 2H), 7.01–6.91 (m, 1H), 5.74 (s, 1H), 5.68 (s, 2H), 3.79 (q, $J$ = 12.0 Hz, 2H), 2.16 (s, 3H), 1.84–1.75 (m, 2H), 1.07 (t, $J$ = 7.4 Hz, 3H); $^{13}$C NMR (100 MHz, CDCl$_3$) $\delta$: 185.0, 170.5, 163.6, 163.1, 157.3 (d, $J$ = 244.0 Hz), 149.7 (d, $J$ = 3.0 Hz), 123.8 (d, $J$ = 10.5 Hz), 117.7 (d, $J$ = 26.1 Hz), 115.7 (d, $J$ = 8.7 Hz), 114.6 (d, $J$ = 22.6 Hz), 107.5, 95.9, 64.8, 48.0, 23.0, 19.9, 11.4; HRMS: calcd for C$_{17}$H$_{18}$ClFNO$_4$$^+$ [M + H]$^+$ 354.0903, found 354.0915; calcd for C$_{17}$H$_{17}$ClFNNaO$_4$$^+$ [M + Na]$^+$ 376.0722, found 376.0732.

Compound 3-(1-(Butylamino)-2-(2-chloro-4-fluorophenoxy)ethylidene)-6-methyl-2$H$-pyran-2,4(3$H$)-dione (**APD-II-3**): white solid; yield 67.0%; $^1$H NMR (400 MHz, CDCl$_3$) $\delta$: 14.43 (s, 1H), 7.17–7.13 (m, 2H), 7.00–6.92 (m, 1H), 5.74 (s, 1H), 5.68 (s, 2H), 3.84–3.79 (m, 2H), 2.15 (s, 3H), 1.78–1.71 (m, 2H), 1.54–1.44 (m, 2H), 0.97 (t, $J$ = 7.3 Hz, 3H); $^{13}$C NMR (100 MHz, CDCl$_3$) $\delta$: 185.0, 170.4, 163.6, 163.1, 157.3 (d, $J$ = 244.2 Hz), 149.7 (d, $J$ = 3.0 Hz), 123.8 (d, $J$ = 10.6 Hz), 117.7 (d, $J$ = 26.2 Hz), 115.8 (d, $J$ = 8.8 Hz), 114.6 (d, $J$ = 22.6 Hz), 107.5, 95.9, 64.8, 46.1, 31.5, 20.1, 19.9, 13.6; HRMS: calcd for C$_{18}$H$_{20}$ClFNO$_4$$^+$ [M + H]$^+$ 368.1059, found 368.1069; calcd for C$_{18}$H$_{19}$ClFNNaO$_4$$^+$ [M + Na]$^+$ 390.0879, found 390.0888.

Compound 3-(2-(2-Chloro-4-fluorophenoxy)-1-(pentylamino)ethylidene)-6-methyl-2$H$-pyran-2,4(3$H$)-dione (**APD-II-4**): white solid; yield 72.3%; $^1$H NMR (400 MHz, CDCl$_3$) $\delta$: 14.43 (s, 1H), 7.18–7.11 (m, 2H), 6.99–6.92 (m, 1H), 5.75 (s, 1H), 5.68 (s, 2H), 3.83–3.78 (m, 2H), 2.15 (s, 3H), 1.85–1.67 (m, 2H), 1.51–1.31 (m, 4H), 0.92 (t, $J$ = 7.0 Hz, 3H); $^{13}$C NMR (100 MHz, CDCl$_3$) $\delta$: 185.0, 170.4, 163.6, 163.1, 157.3 (d, $J$ = 244.0 Hz), 149.7 (d, $J$ = 3.0 Hz), 123.8 (d, $J$ = 10.5 Hz), 117.7 (d, $J$ = 26.1 Hz), 115.8 (d, $J$ = 8.8 Hz), 114.6 (d, $J$ = 22.6 Hz), 107.5, 95.9, 64.7, 46.4, 29.3, 28.9, 22.3, 19.9, 13.9; HRMS: calcd for C$_{19}$H$_{22}$ClFNO$_4$$^+$ [M + H]$^+$ 382.1216, found 382.1227; calcd for C$_{19}$H$_{21}$ClFNNaO$_4$$^+$ [M + Na]$^+$ 404.1035, found 404.1044.

Compound 3-(2-(2-Chloro-4-fluorophenoxy)-1-(hexylamino)ethylidene)-6-methyl-2$H$-pyran-2,4(3$H$)-dione (**APD-II-5**): white solid; yield 70.0%; $^1$H NMR (400 MHz, CDCl$_3$) $\delta$: 14.42 (s, 1H), 7.20–7.09 (m, 2H), 7.03–6.88 (m, 1H), 5.74 (s, 1H), 5.68 (s, 2H), 3.83–3.78 (m, 2H), 2.15 (s, 3H), 1.80–1.70 (m, 2H), 1.51–1.40 (m, 2H), 1.36–1.26 (m, 4H), 0.89 (t, $J$ = 6.7 Hz, 3H); $^{13}$C NMR (100 MHz, CDCl$_3$) $\delta$: 185.0, 170.4, 163.6, 163.1, 157.3 (d, $J$ = 244.0 Hz), 149.7 (d, $J$ = 2.9 Hz), 123.8 (d, $J$ = 10.4 Hz), 117.7 (d, $J$ = 26.1 Hz), 115.8 (d, $J$ = 8.8 Hz), 114.6 (d, $J$ = 22.7 Hz), 107.5, 95.9, 64.8, 46.4, 31.3, 29.5, 26.5, 22.5, 19.9, 14.0; HRMS: calcd for C$_{20}$H$_{24}$ClFNO$_4$$^+$ [M + H]$^+$ 396.1372, found 396.1385; calcd for C$_{20}$H$_{23}$ClFNNaO$_4$$^+$ [M + Na]$^+$ 418.1192, found 418.1201.

Compound 3-(2-(2-Chloro-4-fluorophenoxy)-1-(heptylamino)ethylidene)-6-methyl-2$H$-pyran-2,4(3$H$)-dione (**APD-II-6**): white solid; yield 71.1%; $^1$H NMR (400 MHz, CDCl$_3$) $\delta$: 14.42 (s, 1H), 7.17–7.13 (m, 2H), 7.00–6.90 (m, 1H), 5.74 (s, 1H), 5.68 (s, 2H), 3.83–3.78 (m, 2H), 2.15 (s, 3H), 1.81–1.70 (m, 2H), 1.48–1.39 (m, 2H), 1.36–1.19 (m, 6H), 0.88 (t, $J$ = 6.8 Hz, 3H); $^{13}$C NMR (100 MHz, CDCl$_3$) $\delta$: 185.0, 170.4, 163.6, 163.1, 157.3 (d, $J$ = 243.9 Hz), 149.7 (d, $J$ = 3.0 Hz), 123.8 (d, $J$ = 10.7 Hz), 117.7 (d, $J$ = 26.1 Hz), 115.8 (d, $J$ = 8.8 Hz),

114.6 (d, *J* = 22.7 Hz), 107.5, 95.9, 64.6, 46.4, 31.8, 29.6, 29.1, 29.1, 26.8, 22.6, 19.9, 14.1; HRMS: calcd for $C_{21}H_{26}ClFNO_4^+$ [M + H]$^+$ 410.1529, found 410.1542; calcd for $C_{21}H_{25}ClFNNaO_4^+$ [M + Na]$^+$ 432.1348, found 432.1357.

Compound 3-(2-(2-Chloro-4-fluorophenoxy)-1-(isopropylamino)ethylidene)-6-methyl-2*H*-pyran-2,4(3*H*)-dione (**APD-II-7**): white solid; yield 62.3%; $^1$H NMR (400 MHz, CDCl$_3$) *δ*: 14.50 (s, 1H), 7.20–7.13 (m, 2H), 6.99–6.94 (m, 1H), 5.74 (s, 1H), 5.65 (s, 2H), 4.58–4.55 (m, 2H), 2.15 (s, 3H), 1.41 (s, 3H), 1.40 (s, 3H); $^{13}$C NMR (100 MHz, CDCl$_3$) *δ*: 185.1, 168.5, 163.7, 163.0, 157.4 (d, *J* = 244.3 Hz), 149.9 (d, *J* = 2.8 Hz), 123.9 (d, *J* = 10.7 Hz), 117.7 (d, *J* = 26.3 Hz), 116.0 (d, *J* = 8.7 Hz), 114.6 (d, *J* = 22.6 Hz), 107.5, 95.5, 64.6, 48.3, 23.8, 19.9; HRMS: calcd for $C_{17}H_{18}ClFNO_4^+$ [M + H]$^+$ 354.0903, found 354.0916; calcd for $C_{17}H_{17}ClFNNaO_4^+$ [M + Na]$^+$ 376.0722, found 376.0732.

Compound 3-(1-(t-Butylamino)-2-(2-chloro-4-fluorophenoxy)ethylidene)-6-methyl-2*H*-pyran-2,4(3*H*)-dione (**APD-II-8**): white solid; yield 54.1%; $^1$H NMR (400 MHz, DMSO-*d$_6$*) *δ*: 7.36 (dd, *J* = 8.4, 3.1 Hz, 1H), 7.05 (td, *J* = 8.8, 3.1 Hz, 1H), 6.72 (dd, *J* = 9.2, 5.0 Hz, 1H), 5.45 (s, 1H), 5.15 (s, 2H), 1.98 (s, 3H), 1.17 (s, 9H); $^{13}$C NMR (100 MHz, DMSO-*d$_6$*) *δ*: 190.3, 181.5, 164.5, 160.0, 155.6 (d, *J* = 238.1 Hz), 151.5 (d, *J* = 2.5 Hz), 121.5 (d, *J* = 10.8 Hz), 117.2 (d, *J* = 26.2 Hz), 114.7 (d, *J* = 11.0 Hz), 114.5 (d, *J* = 2.5 Hz), 109.0, 100.6, 74.5, 51.1, 27.4, 19.6; HRMS: calcd for $C_{18}H_{20}ClFNO_4^+$ [M + H]$^+$ 368.1059, found 368.1069; calcd for $C_{18}H_{19}ClFNNaO_4^+$ [M + H]$^+$ 390.0879, found 390.0888.

Compound 3-(2-(2-Chloro-4-fluorophenoxy)-1-(cyclopropylamino)ethylidene)-6-methyl-2*H*-pyran-2,4(3*H*)-dione (**APD-II-9**): white solid; yield 63.2%; $^1$H NMR (400 MHz, CDCl$_3$) *δ*: 14.41 (s, 1H), 7.19–7.09 (m, 2H), 6.96 (td, *J* = 8.8, 3.0 Hz, 1H), 5.76 (s, 2H), 5.72 (s, 1H), 2.15 (s, 3H), 1.05 (q, *J* = 6.9 Hz, 2H), 0.88 (q, *J* = 7.4 Hz, 2H); $^{13}$C NMR (100 MHz, CDCl$_3$) *δ*: 185.0, 172.0, 157.3 (d, *J* = 243.9 Hz), 149.9 (d, *J* = 2.9 Hz), 123.9 (d, *J* = 10.5 Hz), 117.7 (d, *J* = 26.1 Hz), 115.7 (d, *J* = 8.8 Hz), 114.5 (d, *J* = 22.7 Hz), 107.4, 96.1, 65.0, 27.8, 19.9, 8.4; HRMS: calcd for $C_{17}H_{16}ClFNO_4^+$ [M + H]$^+$ 352.0746, found 352.0757; calcd for $C_{17}H_{15}ClFNNaO_4^+$ [M + Na]$^+$ 374.0566, found 374.0575.

Compound 3-(2-(2-Chloro-4-fluorophenoxy)-1-(cyclopentylamino)ethylidene)-6-methyl-2*H*-pyran-2,4(3*H*)-dione (**APD-II-10**): white solid; yield 57.7%; $^1$H NMR (400 MHz, CDCl$_3$) *δ*: 14.63 (s, 1H), 7.20–7.08 (m, 2H), 6.96 (td, *J* = 8.9, 3.0 Hz, 1H), 5.73 (s, 1H), 5.67 (s, 2H), 4.72–4.61 (m, 1H), 2.15–2.12 (m, 5H), 1.87–1.83 (m, 2H), 1.78–1.64 (m, 4H); $^{13}$C NMR (100 MHz, CDCl$_3$) *δ*: 185.0, 168.8, 163.6, 163.0, 157.3 (d, *J* = 244.0 Hz), 149.9 (d, *J* = 2.9 Hz), 123.9 (d, *J* = 10.4 Hz), 117.7 (d, *J* = 26.2 Hz), 116.0 (d, *J* = 8.7 Hz), 114.6 (d, *J* = 22.7 Hz), 107.5, 95.6, 64.8, 57.4, 34.3, 23.8, 19.8; HRMS: calcd for $C_{19}H_{20}ClFNO_4^+$ [M + H]$^+$ 380.1059, found 380.1064; calcd for $C_{19}H_{19}ClFNNaO_4^+$ [M + Na]$^+$ 402.0879, found 402.0882.

Compound 3-(2-(2-Chloro-4-fluorophenoxy)-1-(cyclohexylamino)ethylidene)-6-methyl-2*H*-pyran-2,4(3*H*)-dione (**APD-II-11**): white solid; yield 50.3%; $^1$H NMR (400 MHz, CDCl$_3$) *δ*: 14.55 (bs, 1H), 7.21 (dd, *J* = 9.1, 4.8 Hz, 1H), 7.14 (dd, *J* = 7.9, 3.0 Hz, 1H), 6.96 (td, *J* = 9.0, 3.0 Hz, 1H), 5.74 (s, 1H), 5.66 (s, 2H), 4.28–4.19 (m, 1H), 2.15 (s, 3H), 2.02–1.99 (m, 2H), 1.83–1.80 (m, 2H) (dd, *J* = 9.1, 3.9 Hz, 2H), 1.65–1.25 (m, 6H); $^{13}$C NMR (100 MHz, CDCl$_3$) *δ*: 185.1, 168.5, 163.7, 163.0, 157.4 (d, *J* = 244.1 Hz), 149.9 (d, *J* = 2.9 Hz), 123.9 (d, *J* = 10.5 Hz), 117.7 (d, *J* = 26.1 Hz), 116.3 (d, *J* = 8.8 Hz), 114.6 (d, *J* = 22.7 Hz), 107.5, 95.6, 64.6, 54.8, 33.6, 25.0, 24.1, 19.8; HRMS: calcd for $C_{20}H_{22}ClFNO_4^+$ [M + H]$^+$ 394.1216, found 394.1230; calcd for $C_{20}H_{21}ClFNNaO_4^+$ [M + Na]$^+$ 416.1035, found 416.1045.

Compound 3-(2-(2-Chloro-4-fluorophenoxy)-1-(phenylamino)ethylidene)-6-methyl-2*H*-pyran-2,4(3*H*)-dione (**APD-II-12**): white solid; yield 40.0%; $^1$H NMR (400 MHz, CDCl$_3$) *δ*: 16.09 (s, 1H), 7.43–7.32 (m, 5H), 7.08 (dd, *J* = 8.0, 2.9 Hz, 1H), 6.99 (dd, *J* = 9.1, 4.9 Hz, 1H), 6.90 (td, *J* = 8.0, 4.9 Hz, 1H), 5.83 (s, 1H), 5.34 (s, 2H), 2.19 (s, 3H); $^{13}$C NMR (100 MHz, CDCl$_3$) *δ*: 185.7, 168.4, 164.2, 162.6, 157.2 (d, *J* = 243.6 Hz), 150.1 (d, *J* = 2.9 Hz), 136.2, 129.6, 128.4, 125.1, 124.0 (d, *J* = 10.5 Hz), 117.5 (d, *J* = 26.1 Hz), 115.7 (d, *J* = 8.8 Hz), 114.2 (d, *J* = 22.7 Hz), 107.2, 97.1, 64.8, 20.1; HRMS: calcd for $C_{20}H_{16}ClFNO_4^+$ [M + H]$^+$ 388.0746, found 388.0755; calcd for $C_{20}H_{15}ClFNNaO_4^+$ [M + Na]$^+$ 410.0566, found 410.0573.

Compound 3-(2-(2-Chloro-4-fluorophenoxy)-1-(*p*-tolylamino)ethylidene)-6-methyl-2*H*-pyran-2,4(3*H*)-dione (**APD-II-13**): white solid; yield 42.8%; $^1$H NMR (400 MHz, CDCl$_3$)

$\delta$: 15.99 (s, 1H), 7.25 (d, *J* = 9.1 Hz, 2H), 7.20 (d, *J* = 8.4 Hz, 2H), 7.09 (dd, *J* = 8.0, 3.0 Hz, 1H), 7.00 (dd, *J* = 9.1, 4.9 Hz, 1H), 6.93–6.89 (m, 1H), 5.83 (s, 1H), 5.31 (s, 2H), 2.37 (s, 3H), 2.18 (s, 3H); $^{13}$C NMR (100 MHz, CDCl$_3$) $\delta$: 185.6, 168.2, 162.7, 150.2, 138.6, 133.4, 130.1, 124.9, 124.0 (d, *J* = 10.7 Hz), 117.5 (d, *J* = 26.0 Hz), 115.7 (d, *J* = 8.9 Hz), 114.2 (d, *J* = 22.7 Hz), 107.2, 101.4, 99.9, 97.1, 64.8, 21.1, 20.1; HRMS: calcd for C$_{21}$H$_{18}$ClFNO$_4^+$ [M + H]$^+$ 402.0903, found 402.0912; calcd for C$_{21}$H$_{17}$ClFNNaO$_4^+$ [M + Na]$^+$ 424.0722, found 424.0729.

Compound 3-(2-(2-Chloro-4-fluorophenoxy)-1-((4-fluorophenyl)amino)ethylidene)-6-methyl-2*H*-pyran-2,4(3*H*)-dione (**APD-II-14**): white solid; yield 35.5%; $^1$H NMR (400 MHz, CDCl$_3$) $\delta$: 16.06 (s, 1H), 7,37–7.33 (m, 2H), 7.13–7.07 (m, 3H), 7.00 (dd, *J* = 9.1, 4.8 Hz, 1H), 6.95–6.86 (m, 1H), 5.84 (s, 1H), 5.34 (s, 2H), 2.20 (s, 3H); $^{13}$C NMR (100 MHz, CDCl$_3$) $\delta$: 185.7, 168.8, 164.4, 162.6, 162.1 (d, *J* = 244.2 Hz), 160.9, 157.3 (d, *J* = 243.9 Hz), 149.9 (d, *J* = 2.9 Hz), 132.3, 126.9 (d, *J* = 8.7 Hz), 123.9 (d, *J* = 10.5 Hz), 117.6 (d, *J* = 26.1 Hz), 116.6 (d, *J* = 23.0 Hz), 115.7 (d, *J* = 8.8 Hz), 114.3 (d, *J* = 22.7 Hz), 97.2, 64.8, 20.1; HRMS: calcd for C$_{20}$H$_{15}$ClF$_2$NO$_4^+$ [M + H]$^+$ 406.0652, found 406.0659; calcd for C$_{20}$H$_{14}$ClF$_2$NNaO$_4^+$ [M + Na]$^+$ 428.0472, found 428.0474.

Compound 3-(2-(2-Chloro-4-fluorophenoxy)-1-((2-hydroxyethyl)amino)ethylidene)-6-methyl-2*H*-pyran-2,4(3*H*)-dione (**APD-II-15**): white solid; yield 70.1%; $^1$H NMR (400 MHz, CDCl$_3$) $\delta$: 14.49 (s, 1H), 7.18–7.10 (m, 2H), 7.01–6.88 (m, 1H), 5.72 (s, 3H), 3.99–3.93 (m, 4H), 2.14 (s, 3H); $^{13}$C NMR (100 MHz, CDCl$_3$) $\delta$: 184.9, 171.1, 163.5, 163.4, 157.3 (d, *J* = 244.2 Hz), 149.6 (d, *J* = 2.9 Hz), 123.7 (d, *J* = 10.4 Hz), 117.8 (d, *J* = 26.2 Hz), 115.6 (d, *J* = 8.8 Hz), 114.6 (d, *J* = 22.7 Hz), 107.6, 96.2, 64.9, 60.8, 48.4, 19.8; HRMS: calcd for C$_{16}$H$_{16}$ClFNO$_5^+$ [M + H]$^+$ 356.0696, found 356.0710; calcd for C$_{16}$H$_{15}$ClFNNaO$_5^+$ [M + Na]$^+$ 378.0515, found 378.0526.

Compound 3-(2-(2-Chloro-4-fluorophenoxy)-1-((1-hydroxypropan-2-yl)amino)ethylidene)-6-methyl-2*H*-pyran-2,4(3*H*)-dione (**APD-II-16**): white solid; yield 56.3%; $^1$H NMR (400 MHz, CDCl$_3$) $\delta$: 14.33 (s, 1H), 7.17–7.12 (m, 2H), 6.99–6.94 (m, 1H), 5.91 (d, *J* = 11.2 Hz, 1H), 5.67 (s, 1H), 5.48 (d, *J* = 12.3 Hz, 1H), 4.57–4.52 (m, 1H), 3.86 (dd, *J* = 11.5, 3.5 Hz, 1H), 3.66 (dd, *J* = 11.6, 7.2 Hz, 1H), 2.13 (s, 3H), 1.37 (d, *J* = 6.6 Hz, 3H); $^{13}$C NMR (100 MHz, CDCl3) $\delta$: 185.0, 170.2, 163.7, 163.5, 157.3 (d, *J* = 244.2 Hz), 149.8 (d, *J* = 3.0 Hz), 123.7 (d, *J* = 10.4 Hz), 117.7 (d, *J* = 26.2 Hz), 115.8 (d, *J* = 8.8 Hz), 114.6 (d, *J* = 22.6 Hz), 107.3, 95.8, 66.5, 64.7, 54.3, 19.8, 17.6; HRMS: calcd for C$_{17}$H$_{18}$ClFNO$_5^+$ [M + H]$^+$ 370.0852, found 370.0872; calcd for C$_{17}$H$_{17}$ClFNNaO$_5^+$ [M + Na]$^+$ 392.0671, found 392.0687.

Compound 3-(1-(Benzylamino)-2-(2-chloro-4-fluorophenoxy)ethylidene)-6-methyl-2*H*-pyran-2,4(3*H*)-dione (**APD-II-17**): white solid; yield 66.1%; $^1$H NMR (400 MHz, CDCl$_3$) $\delta$: 14.75 (s, 1H), 7.43–7.32 (m, 5H), 7.22–7.11 (m, 2H), 7.03–6.91 (m, 1H), 5.73 (s, 1H), 5.71 (s, 2H), 5.04 (d, *J* = 5.6 Hz, 2H), 2.15 (s, 3H); $^{13}$C NMR (100 MHz, CDCl$_3$) $\delta$: 185.1, 170.6, 163.5, 163.3, 157.4 (d, *J* = 244.4 Hz), 149.6 (d, *J* = 3.0 Hz), 135.2, 129.2, 128.4, 127.7, 124.0 (d, *J* = 10.5 Hz), 117.8 (d, *J* = 26.2 Hz), 115.9 (d, *J* = 8.9 Hz), 114.7 (d, *J* = 22.7 Hz), 107.5, 96.4, 65.0, 50.0, 19.9; HRMS: calcd for C$_{21}$H$_{18}$ClFNO$_4^+$ [M + H]$^+$ 402.0903, found 402.0916; calcd for C$_{21}$H$_{17}$ClFNNaO$_4^+$ [M + Na]$^+$ 424.0722, found 424.0733.

Compound 3-(2-(2-Chloro-4-fluorophenoxy)-1-(phenethylamino)ethylidene)-6-methyl-2*H*-pyran-2,4(3*H*)-dione (**APD-II-18**): white solid; yield 69.6%; $^1$H NMR (400 MHz, CDCl$_3$) $\delta$: 14.49 (s, 1H), 7.34–7.31 (m, 2H), 7.28–7.22 (m, 3H), 7.13 (dd, *J* = 7.9, 3.0 Hz, 1H), 7.08 (dd, *J* = 9.1, 4.8 Hz, 1H), 6.96–6.92 (m, 1H), 5.74 (s, 1H), 5.53 (s, 2H), 4.11–4.06 (m, 2H), 3.06 (t, *J* = 7.1 Hz, 1H), 2.14 (s, 3H); $^{13}$C NMR (100 MHz, CDCl$_3$) $\delta$: 185.0, 170.7, 163.5, 163.1, 157.4 (d, *J* = 244.2 Hz), 149.7 (d, *J* = 2.9 Hz), 137.3, 128.9 (d, *J* = 8.6 Hz), 127.2, 123.8 (d, *J* = 10.5 Hz), 117.7 (d, *J* = 26.2 Hz), 115.8 (d, *J* = 8.9 Hz), 114.6 (d, *J* = 22.7 Hz), 107.5, 96.0, 64.7, 47.7, 36.1, 19.9; HRMS: calcd for C$_{22}$H$_{20}$ClFNO$_4^+$ [M + H]$^+$ 416.1059, found 416.1071; calcd for C$_{22}$H$_{19}$ClFNNaO$_4^+$ [M + Na]$^+$ 438.0879, found 438.0889.

Compound 3-(2-(2-Chloro-4-fluorophenoxy)-1-((1-phenylethyl)amino)ethylidene)-6-methyl-2*H*-pyran-2,4(3*H*)-dione (**APD-II-19**): white solid; yield 48.6%; $^1$H NMR (400 MHz, CDCl$_3$) $\delta$: 14.93 (s, 1H), 7.39–7.28 (m, 5H), 7.16–7.07 (m, 2H), 6.98–6.89 (m, 1H), 6.03 (d, *J* = 12.7 Hz, 1H), 5.77 (s, 1H), 5.68–5.52 (m, 1H), 4.99 (d, *J* = 12.8 Hz, 1H), 2.15 (s, 3H), 1.71 (d, *J* = 6.8 Hz, 3H); $^{13}$C NMR (100 MHz, CDCl$_3$) $\delta$: 185.3, 169.5, 163.5, 163.3, 157.4 (d, *J* = 244.3 Hz), 149.8 (d, *J* = 2.8 Hz), 141.5, 129.2, 128.1, 125.9, 124.0 (d, *J* = 10.5 Hz),

117.7 (d, *J* = 26.1 Hz), 116.3 (d, *J* = 8.8 Hz), 114.6 (d, *J* = 22.7 Hz), 107.5, 96.2, 64.9, 55.6, 24.4, 19.9; HRMS: calcd for $C_{22}H_{20}ClFNO_4^+$ $[M + H]^+$ 416.1059, found 416.1071; calcd for $C_{22}H_{19}ClFNNaO_4+$ $[M + Na]^+$ 438.0879, found 438.0889.

### 2.3. X-ray Diffraction Analysis of Target Compound APD-I-9

The single crystal of **APD-I-9** was slowly cultivated from a mixture of dichloromethane and methanol (1/1 by volume). Crystallographic data of compound **APD-I-9** had been deposited in the Cambridge Crystallographic Data Centre as supplementary publications with the deposition number 2210257. The detail data can be acquired free of charge from http://www.ccdc.cam.ac.uk/. (accessed on 29 September 2022)

### 2.4. Evaluation of Herbicidal Activity

Herbicidal activity was evaluated based on the reported methods [34–37]. The commercial herbicide atrazine was selected as the positive control. The herbicidal activity of target compounds against two dicotyledonous species: *Abutilon theophrasti Medicus* (AM) and *Amaranthus retroflexus L.* (AR), and two monocotyledonous species: *Echinochloa crus-galli* (EC) and *Digitaria sanguinalis (L.) Scop.* (DS) was tested in a greenhouse. *Poa annua L.* (PA), *Eleusine indica* (EI), *Eragrostis curvula (Schrad.) Nees* (EN), *Avena fatua L.* (AF), *Setaria viridis (L.) Beauv.* (SB), *Cyperus difformis L.* (CD), *Leptochloa chinensis (L.) Nees* (LC), *Chenopodium album L.* (CA), *Ixeris denticulata* (ID), *Plantago asiatica L.* (PA), *Capsella bursa-pastoris (Linn.) Medic* (CB) and *Flaveria bidentis (L.) Kuntze* (FB) were also evaluated in herbicidal spectrum activity experiments. All of the target compounds were first dissolved in DMF and then diluted with Tween-80 to 100 g/L. The solutions were diluted with water to the appropriate concentrations and sprayed immediately after seed planting (pre-emergence treatment) or after the expansion of the first true leaf (post-emergence treatment). A mixture of the same amount of water, *N*, *N*-dimethylformamide and Tween 80 was sprayed as the control. The fresh weight of the above ground tissues was measured 14 days (pre-emergence treatment) or 21 days (post-emergence treatment) after treatment. The inhibition percent was used to describe the control efficiency of the compounds. The data represented the percentage displaying herbicidal damage as compared to the control, where complete control of the target is 100 and no control is 0. All of the bioassays were tested for three parallel experiments. SPSS 22.0 (SPSS, Chicago, IL, USA) was used as the statistical software program. Details of the experimental procedure are given in Supporting Information.

### 2.5. Crop Selectivity

The crops, i.e., wheat, soybean, millet, maize, cotton, peanut and sorghum, were planted in flowerpots (12 cm diameter) and grown at room temperature in the test soil. Crop safety experiments were conducted at a dosage of 375 g ha$^{-1}$ when the crops had reached the four-leaf stage. After 14 days of treatment by compound **APD-II-15**, the crop selectivity was evaluated with three duplicates per experiment. The data represented the percentage displaying damage as compared to the control, where complete control of the target is 100 and no control is 0.

### 2.6. Phenotypic Study of Arabidopsis Thaliana

The *Arabidopsis thaliana* employed in this study is in the Columbia (Col-0) background. Surface-sterilized seeds were sown onto Murashige and Skoog (MS) plates at 4 °C for 3 days under darkness to favor vernalization and then grown at 22 °C for 2 days under LD (16 h light/8 h darkness). Subsequently, the plants were transferred to 1/2 MS plates supplemented with compound **APD-II-15** (0.01 μM) and atrazine (0.01 μM), respectively. The solution of compound **APD-II-15** and atrazine was prepared in 0.1% DMF and diluted in the 1/2 MS to obtain the tested concentrations, respectively. The solvent (0.1% DMF) was also added to control plates. The plants were grown on vertically oriented plates at 22 °C (air humidity of 40–60%) for 7 days under LD (16 h light/8 h darkness) for phenotypic investigation.

### 2.7. RNA Sequencing and Data Analysis

*A.thaliana* was cultured in an incubator with a temperature of 22 °C and a 16/8 h light/ dark photoperiod for 14 days. The compound APD-II-15 was dissolved in DMF and diluted in the deionized water to obtain the tested concentrations. *A. thaliana* seedlings with good growth were selected to be placed in a petri dish (filter-paper covered) and treated with **APD-II-15** (10 mM) as the experimental group. The distilled water (0.1% DMF) was added to the control plates. All treatments were conducted with three replicates. After treatment for 12 h, 0.1 g of fresh *A. thaliana* seedlings were weighed into a 15 mL centrifuge tube, quickly frozen and stored in a −80 °C refrigerator. Total RNA extraction and RNA-Seq analysis was performed by Beijing Novogene Biotechnology Co., Ltd. For all samples, total RNAs were extracted using a TRIzol reagent (Invitrogen, USA). RNA integrity was assessed using the RNA Nano 6000 Assay Kit of the Bioanalyzer 2100 system (Agilent Technologies, CA, USA). Total RNA was used as input material for the RNA sample preparations. The clustering of the index-coded samples was performed on a cBot Cluster Generation System using TruSeq PE Cluster Kitv3-cBot-HS (Illumia) according to the manufacturer's instructions. Differential expression analysis of two conditions/groups (dose and control) was performed using the DESeq2 R package (1.20.0), and genes with padj < 0.05 | log2FoldChange | >0.0 found by DESeq2 were assigned as differentially expressed. Gene Ontology (GO) and Kyoto Encyclopedia of Genes and Genomes (KEGG) enrichment analyses were implemented by the clusterProfiler R package (3.4.4) [38].

## 3. Results and Discussion

### 3.1. Chemistry

As depicted in Scheme 1, the target compounds **APD-I** and **APD-II** were easily prepared via a five-step synthetic route. Briefly, in the presence of potassium carbonate as a base, phenol **1** was reacted with methyl chloroacetate **2** to give the corresponding methyl aryloxyacetate **3**. The hydrolysis of methyl aryloxyacetate **3** with sodium hydroxide in the methanol/water system afforded the corresponding aryloxyacetyl acid **4**, which was reacted with oxalyl chloride in the presence of a catalytic amount of DMF to give the corresponding aryloxyacetyl chloride derivatives **5**. Subsequently, reacting compound **6** with aryloxyacetyl chloride **5** in the presence of trimethylamine afforded the key intermediate **7**. Finally, target compounds **APD-I** and **APD-II** were synthesized in 35.5–81.3% yield by the condensation of **7** with the corresponding amine in ethanol. The structures of all the target compounds were identified using $^1$H and $^{13}$C NMR spectroscopy, and HRMS. Furthermore, the structure of compound **APD-I-9** was confirmed using X-ray diffraction analysis (CCDC 2210257; Figure 2).

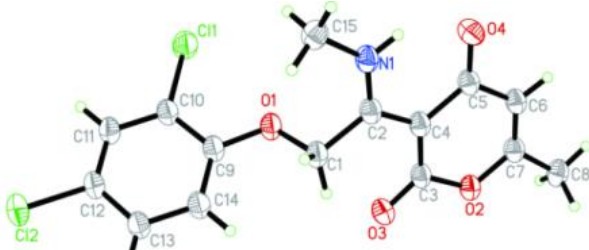

**Figure 2.** X-ray crystal structure of compound **APD-I-9**.

### 3.2. Herbicidal Activity of Target Compounds in Greenhouse Tests and SAR Study

As an initial work, the target compounds **APD-I-1** to **APD-I-14** were first synthesized, and their herbicidal activities were evaluated at a dosage of 1500 g ha$^{-1}$ located in a greenhouse (data are given in Supporting Information, Table S1). Atrazine was selected as the positive control. As shown in Figure 3, some of the target compounds, such as **APD-I-9**, **APD-I-10**, **APD-I-12**, and **APD-I-14**, exhibited >200% sum inhibition against the weeds tested under pre-emergence conditions. Among them, compound **APD-I-14** exhibited

excellent pre-emergent herbicidal activity with a sum inhibition rate 370.3%, which was comparable to the activity observed for atrazine. However, when the herbicidal activity of these target compounds was evaluated under post-emergence conditions, the result was not ideal. Only compound **APD-I-14** exhibited >200% sum inhibition against the weeds tested. It was found that the $R_1$ group on the benzene ring has a significant influence on herbicidal activity. Analyzing the pre-emergent herbicidal activity of these target compounds, it is easy to find that the compounds with a 2,4-di-substitution pattern on the benzene ring, such as **APD-I-9** and **APD-I-12** to **APD-I-14**, exhibited stronger herbicidal activity than that of compounds with mono-pattern or other di-substituted patterns. Among the 2,4-di-substituted analogues, compounds with 2-Cl-4-F substitution on the benzene ring, i.e., **APD-I-14**, possess the highest sum inhibition against the weeds tested, indicating that the 2-Cl-4-F substituent is an optimal orientation and that compound **APD-I-14** can be served as a potential lead compound for further optimization.

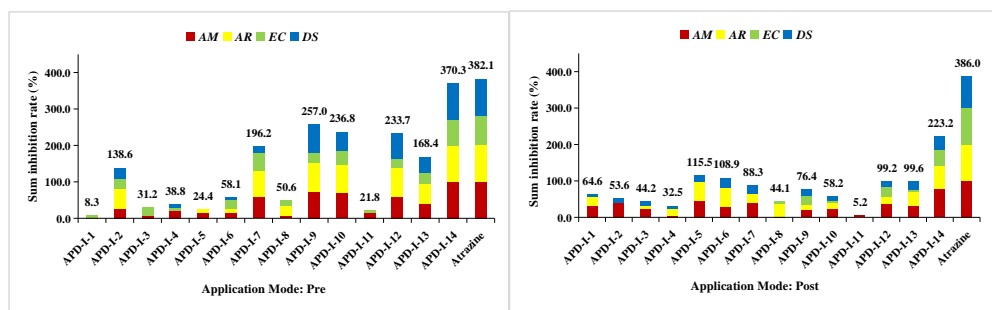

**Figure 3.** Effects (% inhibition) of target compounds **APD-I-1**–**APD-I-14** on loss of plant weight at a dosage of 1500 g ha$^{-1}$ after treatment at 14 days (Pre) or 21 days (Post); Pre—pre-emergence condition; Post—post-emergence condition; *Abutilon theophrasti Medicus* (AM), *Amaranthus retroflexus L.* (AR), *Echinochloa crus-galli* (EC) and *Digitaria sanguinalis (L.) Scop.* (DS).

In order to explore the effect of the substituent at the nitrogen atom of **APD-I-14** on herbicidal activity, the target compounds **APD-II-1** to **APD-II-19** were synthesized and tested for herbicidal activity (data is given in Supporting Information, Table S2). Atrazine and compound **APD-I-14** were selected as positive controls. As shown in Figure 4, the target compounds **APD-II-1** to **APD-II-19** exhibited >300% sum inhibition against the weeds tested at a dosage of 1500 g ha$^{-1}$ under pre-emergence conditions. Among them, compounds **APD-II-15** and **APD-II-16** exhibited excellent pre-emergent herbicidal activity, with sum inhibition rates of 398.1% and 394.8%, respectively, which were even better than that of atrazine. However, the herbicidal activity of compounds **APD-II-1** to **APD-II-19** was decreased under the post-emergence condition, which indicated that the target compounds possess enhanced herbicidal activity under pre-emergence conditions compared to under post-emergence conditions. This promising result encouraged us to further evaluate the herbicidal activity of these target compounds at a lower dosage under pre-emergence conditions.

Subsequently, the herbicidal activity of target compounds **APD-I-14** and **APD-II-1** to **APD-II-19** were tested under pre-emergence conditions by using a dose reduction with serial two-fold dilutions (data are given in Supporting Information, Table S3). With the dosage decreased from 750 g ha$^{-1}$ to 187.5 g ha$^{-1}$ (Figure 5), the herbicidal activity of these target compounds and atrazine became progressively lower. To our delight, when the dosage was decreased to 187.5 g ha$^{-1}$, compound **APD-II-15** still exhibited 270.9% sum inhibition against the tested weeds, which was superior to that of atrazine (sum inhibition rate = 232.8%) (Figure 6). Upon decreasing the dosage, the SAR gradually becomes obvious. It was found that the $R_2$ group at the nitrogen atom also has a significant influence on the herbicidal activity of the target compounds. When analyzing the herbicidal activity of these target compounds at a dosage of 187.5 g ha$^{-1}$ (Figure 5C), it was found that among the compounds with straight-chain alkyl groups (i.e., **APD-I-14**, **APD-II-2** to **APD-II-6**), **APD-II-2** ($R_2$ = n-propyl) exhibited the highest herbicidal activity,

with a sum inhibition rate 196.0%. With the extension of the carbon chain, compounds with a longer carbon chain progressively lost herbicidal activity. When a branched-chain alkyl groups was introduced on the nitrogen atom, these compounds exhibited lower herbicidal activity than that of the corresponding straight-chain compounds. For example, **APD-II-7** ($R_2$ = *i*-propyl, sum inhibition rate = 147%) < **APD-II-2** ($R_2$ = n-propyl, sum inhibition = 196.0%), and **APD-II-8** ($R_2$ = *t*-butyl, sum inhibition rate = 142.2%) < **APD-II-3** ($R_2$ = n-Butyl, sum inhibition = 175.5%). Cycloalkyl-substituted compounds (i.e., **APD-II-9** to **APD-II-11**) also exhibited lower herbicidal activity than that of the corresponding straight-chain compounds, and the herbicidal activity was decreased with the expansion of the ring. With the alkyl group on the nitrogen atom replaced by a benzene ring, compound **APD-II-12** displayed the worst control against the weeds tested. When a methyl or fluorine atom was introduced at the 4-rd of the benzene ring of compound **APD-II-12**, the herbicidal activity of **APD-II-13** and **APD-II-14** was slight improved, which indicated that the electronic effect of the substituent on the benzene ring had a certain influence on the herbicidal activity of the target compounds. The introduction of the alkyl chain between the benzene ring and the nitrogen atom also slightly improved the herbicidal activity, and the herbicidal activity of compounds **APD-II-12** and **APD-II-17** to **APD-II-19** could be placed in the following order: **APD-II-18** ($R_2$ = phenylethyl) > **APD-II-17** ($R_2$ = benzyl) > **APD-II-19** ($R_2$ = 1-phenylethyl) > **APD-II-12** ($R_2$ = phenyl). These findings demonstrate that the introduction of sterically bulky substituents at the at the nitrogen atom of **APD-I-14** is not conducive to improve herbicidal activity.

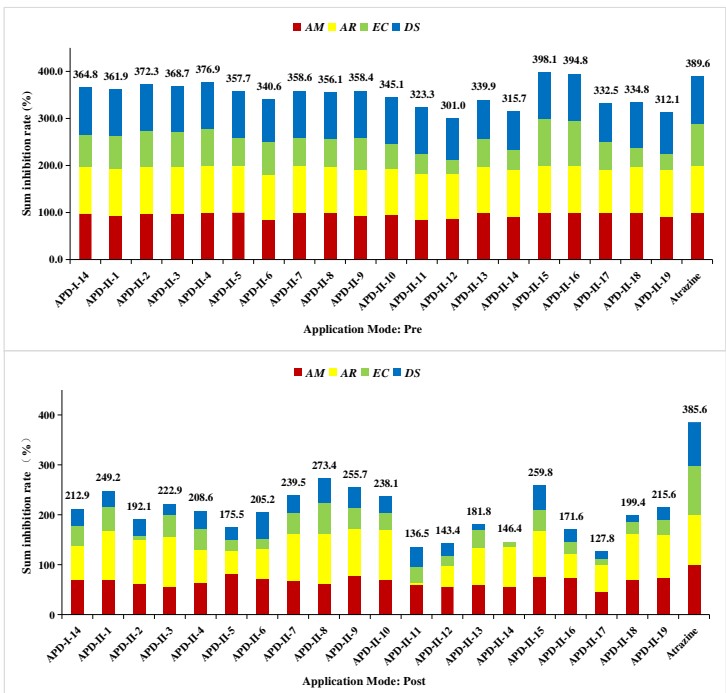

**Figure 4.** Effects (% inhibition) of compounds **APD-I-14**, **APD-II-1**–**APD-II-19** on loss of plant weight at a dosage of 1500 g ha$^{-1}$ after treatment at 14 days (Pre) or 21 days (Post); Pre—pre-emergence condition; Post—post-emergence condition; *Abutilon theophrasti Medicus* (AM), *Amaranthus retroflexus L.* (AR), *Echinochloa crus-galli* (EC) and *Digitaria sanguinalis (L.) Scop.* (DS).

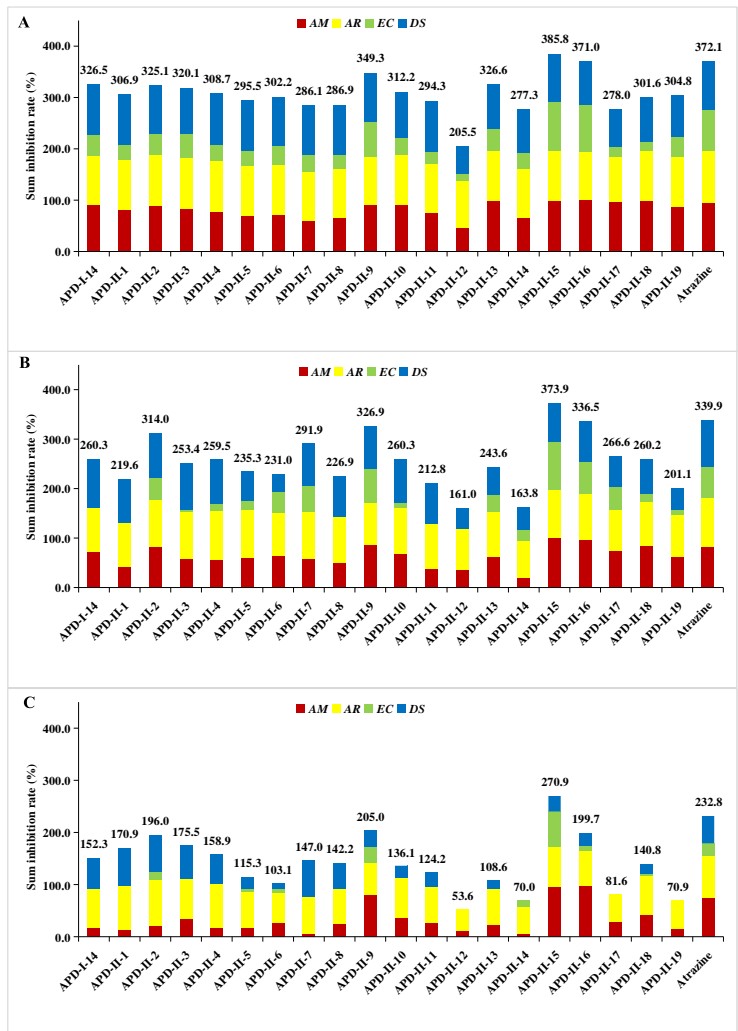

**Figure 5.** Effects (% inhibition) of compounds **APD-I-14**, **APD-II-1** to **APD-II-19** on loss of plant weight at a different dosage under pre-emergence conditions after treatment at 14 days; (**A**) at a dosage of 750 g ha$^{-1}$; (**B**) at a dosage of 375 g ha$^{-1}$; (**C**) at a dosage of 187.5 g ha$^{-1}$. *Abutilon theophrasti Medicus* (AM), *Amaranthus retroflexus L.* (AR), *Echinochloa crus-galli* (EC) *Digitaria sanguinalis (L.) Scop.* (DS).

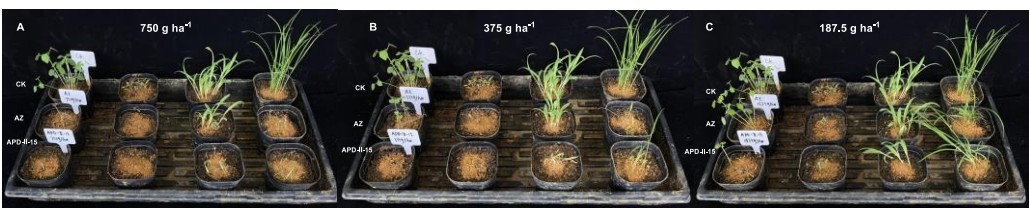

**Figure 6.** Photographs illustrating loss of plant weight at a different dosage of compound **APD-II-15** and atrazine under pre-emergence conditions after treatment at 14 days; (**A**) at a dosage of 750 g ha$^{-1}$; (**B**) at a dosage of 375 g ha$^{-1}$; (**C**) at a dosage of 187.5 g ha$^{-1}$. The top row of each pallet is CK, the middle row is the plant treated by AZ, and the bottom row is the plant treated by APD-II-15. In each pallet, from left to right: *Abutilon theophrasti Medicus* (AM), *Amaranthus retroflexus L.* (AR), *Echinochloa crus-galli* (EC), *Digitaria sanguinalis (L.) Scop.* (DS).

It is noteworthy that introducing a hydroxyl at the terminal of the alkyl chain significantly improved the herbicidal activity (for example, **APD-II-15** (R$_2$ = 2-hydroxyethyl, sum inhibition = 270.9%) > **APD-II-1** (R$_2$ = Et, sum inhibition = 170.9%) and **APD-II-16** (R$_2$ = 1-hydroxypropan-2-yl, sum inhibition = 199.7%) > **APD-II-7** (R$_2$ = i-propyl, sum

inhibition = 147%)), implying that the introduction of hydrophilic groups to the target compound is conducive to improve herbicidal activity. Considering that **APD-II-15** possess the highest sum inhibitory activity against the weeds tested, it was confirmed as a potential herbicide lead compound for further investigation.

### 3.3. Herbicidal Spectrum and Crop Safety of Compound APD-II-15

Since compound **APD-II-15** stood out as the most potential herbicide lead compound, its herbicidal spectrum was investigated under pre-emergence conditions. As shown in Figure 7, as a whole, **APD-II-15** displays strong control with inhibition >60% against 11 of the 16 weeds tested at a dosage of 187.5 g ha$^{-1}$ under pre-emergence conditions, while atrazine displays strong control with inhibition >60% against 9 of the 16 tested weeds under the same conditions. This indicated that compound **APD-II-15** had a broader spectrum of weed control than the commercial herbicide atrazine. It is noteworthy that atrazine has no control against *Eragrostis curvula*, whereas compound **APD-II-15** displayed strong control with >70% inhibition to this weed, indicating that atrazine could be used in combination with compound **APD-II-15** to broaden the herbicidal spectrum. To evaluate whether compound **APD-II-15** has the potential to be developed as an herbicide candidate, the test of crop selectivity was performed at a dosage of 375 g ha$^{-1}$ under pre-emergence conditions. The results shown in Table 1 revealed that wheat, soybean, millet and sorghum possessed a high tolerance toward **APD-II-15**, but **APD-II-15** was not selective for maize (10.7% injury), cotton (37.6% injury) and peanut (26.9% injury). These promising results indicated that compound **APD-II-15** can be served as a potential herbicide candidate for weed control in wheat, soybean, millet and sorghum fields.

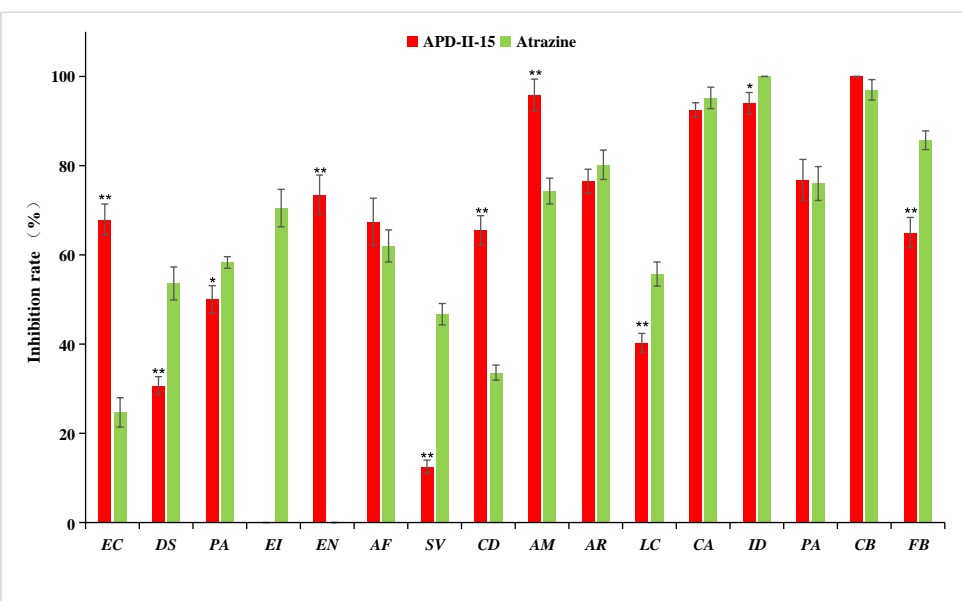

**Figure 7.** Effects (% inhibition) of the target compound **APD-II-15** and atrazine on loss of plant weight at a dosage of 187.5 g ha$^{-1}$ after treatment at 14 days under the pre-emergence conditions. *Abutilon theophrasti Medicus* (AM), *Amaranthus retroflexus L.* (AR), *Echinochloa crus-galli* (EC) *Digitaria sanguinalis (L.) Scop.* (DS), *Poa annua L.* (PA), *Eleusine indica* (EI), *Eragrostis curvula (Schrad.) Nees* (EN), *Avena fatua L.* (AF), *Setaria viridis (L.) Beauv.* (SB), *Cyperus difformis L.* (CD), *Leptochloa chinensis (L.) Nees* (LC), *Chenopodium album L.* (CA), *Ixeris denticulata* (ID), *Plantago asiatica L.* (PA), *Capsella bursa-pastoris (Linn.) Medic* (CB), *Flaveria bidentis (L.) Kuntze* (FB); Asterisk (*) means significant differences between different treatments. *, significant at the *p* < 0.05 level, and **, significant at the *p* < 0.01 level compared with atrazine.

**Table 1.** Pre-emergence crop selectivity of **APD-II-15** at the dosage of 375 g ha$^{-1}$ (Injury inhibition) [a].

| Comp. | Maize | Wheat | Soybean | % Injury Cotton | Millet | Peanut | Sorghum |
|---|---|---|---|---|---|---|---|
| **APD-II-15** | 10.7 ± 1.3 ** | 0 | 0 | 37.6 ± 3.1 ** | 0 | 26.9 ± 1.4 ** | 0 |
| Atrazine | 0 | 24.8 ± 2.6 | 2.8 ± 1.1 | 0 | 16.7 ± 3.4 | 0 | 0 |

[a] Each value represents the mean ± SD of three experiments. Asterisk (*) means significant differences between different treatments. **, significant at the $p < 0.01$ level compared with atrazine.

### 3.4. Evaluation of Molecular Mode of Action of the Target Compounds

In order to explore the molecular mode of action of the target compound, **APD-II-15** was selected to study the herbicidal mechanism with *Arabidopsis thaliana* as a model plant. As shown in Figure 8, **APD-II-15** showed a certain inhibition against the root and seedling of *A. thaliana*, and the *A. thaliana* treated with **APD-II-15** developed a symptom of yellowed leaves (Figure 8), similar to that of atrazine. However, from this phenotype, it is difficult to confirm the molecular mode of action of **APD-II-15**.

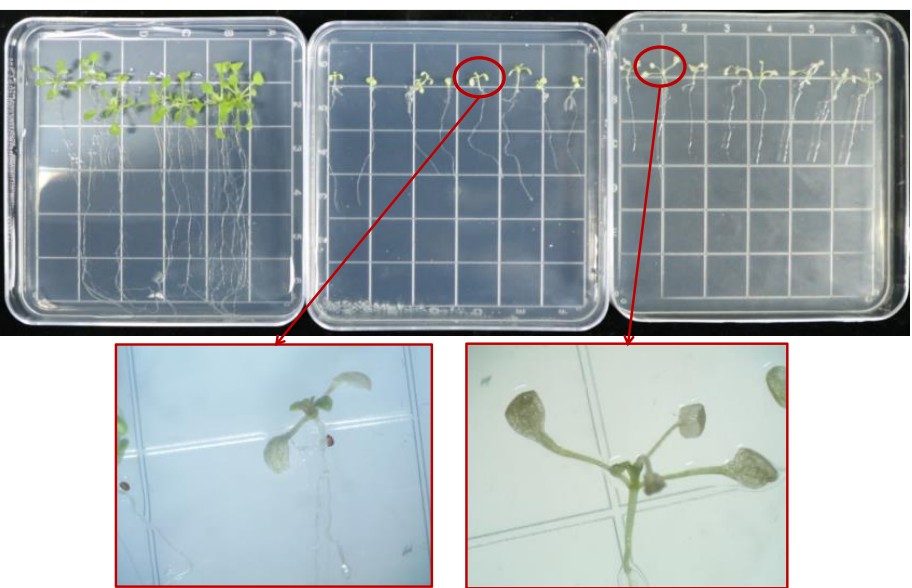

**Figure 8.** Photographs illustrating seedling phenotype of *Arabidopsis thaliana* treated with **APD-II-15** and atrazine.

To understand the molecular mode of action, the RNA sequencing was subsequently performed to identify the differential expression genes (DEGs) of *A. thaliana* treated with compound **APD-II-15**. In total, 16362 DEGs with padj <0.05 | log2FoldChange | >0.0 were identified in **APD-II-15** vs CK. The heatmap and volcano plots for those 16,362 DEGs are illustrated in Figure 9. It was found that there were 7977 up-regulated genes and 8385 downregulated genes. To further understand the functions of those DEGs, GO annotations and KEGG annotations are given in Figure 10. As shown in Figure 10A, GO annotations were performed on the 16,362 DEGs, which were classified into the molecular function, cell component and biological process. Most enriched down-regulated GO terms were related to polysaccharide metabolic processes and cytoskeleton part; meanwhile, KEGG pathway enrichment analysis showed that most DEGs were significantly enriched in the carbon metabolism and glycolysis/gluconeogenesis. These analysis results illustrated that a growth inhibition of weeds by **APD-II-15** might result from disruptions of the formation of the cytoskeleton and carbon metabolism. Further research is still ongoing to clarify the herbicidal mode of action of the target compounds.

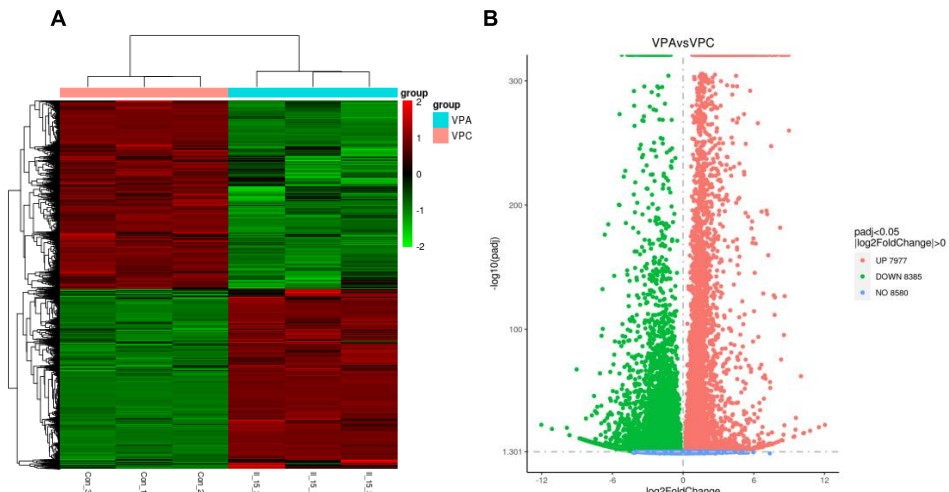

**Figure 9.** RNA sequencing analysis of *A. thaliana* treated by **APD-II-15**. (**A**) Heatmap of differential expression genes; (**B**) volcano plot of differential expression genes.

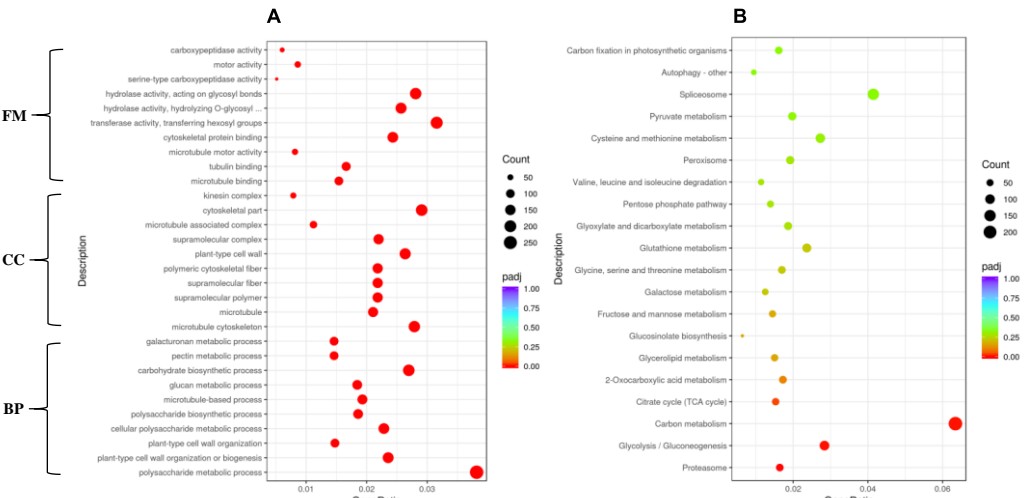

**Figure 10.** RNA sequencing analysis of *A. thaliana* treated by **APD-II-15**. (**A**) GO analysis of differential expression genes (MF: molecular function; CC: cell component; BP: biological process); (**B**) KEGG analysis of differential expression genes.

## 4. Conclusions

In summary, a novel 3-(1-aminoethylidene)-6-methyl-2*H*-pyran-2,4(3*H*)-dione structure has been rationally designed by molecular hybridization between the natural enamino diketone skeleton and the reported herbicide lead compound I; thirty-three **APD** derivatives were prepared in moderate to good yield. After the optimization of the $R_1$ and $R_2$ group, compound **APD-II-15** was confirmed as a potential herbicide lead compound owing to its good herbicidal activity, broad herbicidal spectrum and good crop safety. The investigation of the phenotypes of *A. thaliana* has difficulty in identifying the molecular mode of action of compound **APD-II-15**; thus, RNA sequencing was performed to understand the herbicidal mechanism. The RNA sequencing analysis results suggested that a growth inhibition of weeds by **APD-II-15** might result from the disruptions of the formation of the cytoskeleton and carbon metabolism. The present work demonstrates that compound **APD-II-15** can be served as a potential lead structure for further optimization. Further studies on the structural optimization of compound **APD-II-15** and on the clarifying of the herbicidal mode of action of target compounds are ongoing in our laboratory.

**Supplementary Materials:** The following supporting information can be downloaded at: https: //www.mdpi.com/article/10.3390/agronomy13010202/s1.

**Author Contributions:** Conceptualization, K.L.; methodology, K.L. and K.C.; validation, C.-C.W., N.L.; formal analysis, P.L. and X.-W.H.; investigation, C.-C.W. and N.L.; data curation, X.-K.W., S.-B.W. and X.-W.H.; writing—original draft preparation, K.L.; writing—review and editing, K.L. and K.C.; project administration, K.L.; funding acquisition, K.L., X.-K.W. and L.-S.J. All authors have read and agreed to the published version of the manuscript.

**Funding:** This project was financially supported by the National Natural Science Foundation of China (No. 31701827), the China Postdoctoral Science Foundation (No. 2020M671984), the Guangyue Young Scholar Innovation Team of Liaocheng University (No. LCUGYTD2022-04), the National Key Research and Development Program of China (No. SQ2020YFF0422322) and the Natural Science Foundation of Shandong Province (No. ZR202102180037).

**Institutional Review Board Statement:** Not applicable.

**Informed Consent Statement:** Not applicable.

**Data Availability Statement:** Not applicable.

**Conflicts of Interest:** The authors declare no conflict of interest.

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
