# Peer review of "Discovery of 3-(1-Amino-2-phenoxyethylidene)-6-methyl-2H-pyran-2,4(3H)-dione Derivatives as Novel Herbicidal Leads"

_agronomy, doi:10.3390/agronomy13010202_

Round 1

Reviewer 1 Report

Overall, the work is very well written and easy to follow. I just want to make a few comments in order to improve the manuscript.
- It would be interesting to include a summary table of the compounds prepared at the end of section 2.1, in order to better follow the performed bioassays. It is important to know the structure of the compounds in order to be able to derive SAR conclusions. It is true that the spectroscopic data are at the end of the manuscript, but for the convenience of the reader it would be nice to see the prepared structures.
- I like the representations of bioactivity, although it is more difficult to see which species is more sensitive to which compound. Maybe they could have been put by species/compound, and select only the most active ones, leaving the rest for the complementary material.
- In Figure 5, indicate in the figure caption who is A, B and C.
- In figure 7, which parameter has been evaluated? It is not indicated in the text or in the figure.

Author Response

3, January, 2023

Dear editor,

Thank you very much for your letter. We have revised the paper entitled “Discovery of 3-(1-Amino-2-phenoxyethylidene)-6-methyl-2H-pyran-2,4(3H)-dione Derivatives as Novel Herbicidal Leads” (agronomy-2090823), and would like to resubmit it for your consideration. We have studied reviewer’s comments carefully and addressed the questions raised by the reviewers, and the amendments are highlighted in red in the revised manuscript. We hope that the revision is acceptable, and I look forward to hearing from you soon.

If there are any questions, please contact with me.

Corresponding author: Kang Lei, leikang@lcu.edu.cn

Thank you very much for your attention and consideration.

With best wishes,

Yours sincerely,

Kang Lei

Responses to the comments of reviewers:

Reviewer: 1

  1. It would be interesting to include a summary table of the compounds prepared at the end of section 2.1, in order to better follow the performed bioassays. It is important to know the structure of the compounds in order to be able to derive SAR conclusions. It is true that the spectroscopic data are at the end of the manuscript, but for the convenience of the reader it would be nice to see the prepared structures.

Answer:

Thank you very much for your valuable comment. We have revised the article according to your suggestion, and we have made a summary table of the structure of target compounds at the Scheme 1.

  1. I like the representations of bioactivity, although it is more difficult to see which species is more sensitive to which compound. Maybe they could have been put by species/compound, and select only the most active ones, leaving the rest for the complementary material.

Answer:

Thank you very much for your valuable comment. As we need to analyze the herbicidal activity of the compound as a whole, it is unsuitable to put some of data into the complementary material.

  1. In Figure 5, indicate in the figure caption who is A, B and C.

Answer:

Thank you very much for your valuable comment. We have revised the figure caption according to your suggestion.

  1. In figure 7, which parameter has been evaluated? It is not indicated in the text or in the figure.

Answer:

Thank you very much for your valuable comment. We revised the figure caption to make it easier to understand. 

Reviewer 2 Report

Dear authors,

the paper entitled "Discovery of 3-(1-Amino-2-phenoxyethylidene)-6-methyl-2H-pyran-2,4(3H)-dione Derivatives as Novel Herbicidal Leads" gives insights into a novel herbicidal compound. The paper needs major revision before acceptance. Specifically, the abstract has to be revised and the weeds (or at least some of them) need to be added. The introduction and discussion need substantial changes (see below). The materials and methods section needs only minor changes. However, the addition of statistical analysis is of major importance. The results section is sound.

L34: untreated, not unchecked

L35: efficient weed control method, instead of convenient weed control tool
L38: Mechanism of action, not site of action

L38: evolvement of herbicide resistance, not occurrence

L39-40: This is not the only alternative option. There are many alternative weed management methods and strategies to reduce herbicide input. Authors should cite https://doi.org/10.3390/agronomy12030589 and https://doi.org/10.1016/j.eja.2021.126443

L43: Explain the abbreviation GS

L46: which studies? Please cite the relevant references.

L52: Over the last few decades

L110: When? Add the days after treatment or something relevant. All figure captions need revision.

L178-179: The caption is incomplete. The plants are missing.

DISCUSSION: Add more references comparing the results of your research.

L249: Identify

L258: There is a complete absence of statistical analysis. This is a major point and the authors have to address that.

L599: Name the crops.

Author Response

3, January, 2023

Dear editor,

Thank you very much for your letter. We have revised the paper entitled “Discovery of 3-(1-Amino-2-phenoxyethylidene)-6-methyl-2H-pyran-2,4(3H)-dione Derivatives as Novel Herbicidal Leads” (agronomy-2090823), and would like to resubmit it for your consideration. We have studied reviewer’s comments carefully and addressed the questions raised by the reviewers, and the amendments are highlighted in red in the revised manuscript. We hope that the revision is acceptable, and I look forward to hearing from you soon.

If there are any questions, please contact with me.

Corresponding author: Kang Lei, leikang@lcu.edu.cn

Thank you very much for your attention and consideration.

With best wishes,

Yours sincerely,

Kang Lei

Responses to the comments of reviewers:

  1. L34: untreated, not unchecked

Answer:

Thank you very much for your valuable comment. Suggested revision was performed.

  1. L35: efficient weed control method, instead of convenient weed control tool

Answer:

Thank you very much for your valuable comment. Suggested revision was performed.

  1. L38: Mechanism of action, not site of action

Answer:

Thank you very much for your valuable comment. Suggested revision was performed.

  1. L38: evolvement of herbicide resistance, not occurrence

Answer:

Thank you very much for your valuable comment. Suggested revision was performed.

  1. L39-40: This is not the only alternative option. There are many alternative weed management methods and strategies to reduce herbicide input. Authors should cite https://doi.org/10.3390/agronomy12030589 and https://doi.org/10.1016/j.eja.2021.126443

Answer:

Thank you very much for your valuable comment. Suggested revision was performed.

  1. L43: Explain the abbreviation GS

Answer:

Thank you very much for your valuable comment. Suggested revision was performed.

  1. L46: which studies? Please cite the relevant references.

Answer:

Thank you very much for your valuable comment. Relevant references were cited.

  1. L52: Over the last few decades

Answer:

Thank you very much for your valuable comment. Suggested revision was performed.

  1. L110: When? Add the days after treatment or something relevant. All figure captions need revision.

Answer:

Thank you very much for your valuable comment. Suggested revision was performed.

  1. L178-179: The caption is incomplete. The plants are missing.

Answer:

Thank you very much for your valuable comment. Suggested revision was performed.

  1. DISCUSSION: Add more references comparing the results of your research.

Answer:

Thank you very much for your valuable comment.

  1. L249: Identify

Answer:

Thank you very much for your valuable comment. Suggested revision was performed.

  1. L258: There is a complete absence of statistical analysis. This is a major point and the authors have to address that.

Answer:

Thank you very much for your valuable comment. We have performed a statistical analysis of the data. The data was given in Supporting Information. 

  1. L599: Name the crops.

Answer:

Thank you very much for your valuable comment. We have name the crops.
